# Genomic sequencing of SARS-CoV-2 in Rwanda reveals the importance of incoming travelers on lineage diversity

Yvan Butera[1,2,3,17], Enatha Mukantwari[4,17], Maria Artesi[3,17], Jeanne d'arc Umuringa[4,17], Áine Niamh O'Toole [5], Verity Hill [5], Stefan Rooke[5], Samuel Leandro Hong [6], Simon Dellicour [6,7], Onesphore Majyambere[4], Sebastien Bontems[8], Bouchra Boujemla[3], Josh Quick[9], Paola Cristina Resende[10,11], Nick Loman[9], Esperance Umumararungu[4], Alice Kabanda[4], Marylin Milumbu Murindahabi[2,12], Patrick Tuyisenge [4], Misbah Gashegu[2], Jean Paul Rwabihama[2], Reuben Sindayiheba[4], Djordje Gikic [2], Jacob Souopgui [1,13], Wilfred Ndifon[14], Robert Rutayisire[2,4], Swaibu Gatare[2,4], Tharcisse Mpunga[2], Daniel Ngamije[2], Vincent Bours[3,15,17], Andrew Rambaut [5,17], Sabin Nsanzimana[2,17], Guy Baele [6,17✉], Keith Durkin [3,17✉], Leon Mutesa [1,2,17✉] & Nadine Rujeni [2,16,17✉]

COVID-19 transmission rates are often linked to locally circulating strains of SARS-CoV-2. Here we describe 203 SARS-CoV-2 whole genome sequences analyzed from strains circulating in Rwanda from May 2020 to February 2021. In particular, we report a shift in variant distribution towards the emerging sub-lineage A.23.1 that is currently dominating. Furthermore, we report the detection of the first Rwandan cases of the B.1.1.7 and B.1.351 variants of concern among incoming travelers tested at Kigali International Airport. To assess the importance of viral introductions from neighboring countries and local transmission, we exploit available individual travel history metadata to inform spatio-temporal phylogeographic inference, enabling us to take into account infections from unsampled locations. We uncover an important role of neighboring countries in seeding introductions into Rwanda, including those from which no genomic sequences were available. Our results highlight the importance of systematic genomic surveillance and regional collaborations for a durable response towards combating COVID-19.

---

[1] Center for Human Genetics, College of Medicine and Health Sciences, University of Rwanda, Kigali, Rwanda. [2] Rwanda National Joint Task Force COVID-19, Rwanda Biomedical Centre, Ministry of Health, Kigali, Rwanda. [3] Laboratory of Human Genetics, GIGA Research Institute, Liège, Belgium. [4] National Reference Laboratory, Rwanda Biomedical Center, Kigali, Rwanda. [5] Institute of Evolutionary Biology, University of Edinburgh, Edinburgh, Scotland. [6] Department of Microbiology, Immunology and Transplantation, Rega Institute KU Leuven, Leuven, Belgium. [7] Spatial Epidemiology Laboratory, Université Libre de Bruxelles, Brussels, Belgium. [8] Department of Clinical Microbiology, University Hospital of Liège, Liège, Belgium. [9] University of Birmingham, Birmingham, England. [10] University College London, London, England. [11] Laboratory of Respiratory Viruses and Measles, Oswaldo Cruz Institute, FIOCRUZ, Rio de Janeiro, Brazil. [12] School of Science, College of Science and Technology, University of Rwanda, Kigali, Rwanda. [13] Department of Molecular Biology, Institute of Biology and Molecular Medicine, IBMM, Université Libre de, Bruxelles, Gosselies, Belgium. [14] African Institute for Mathematical Sciences, Kigali, Rwanda. [15] Department of Human Genetics, University Hospital of Liège, Liège, Belgium. [16] School of Health Sciences, College of Medicine and Health Sciences, University of Rwanda, Kigali, Rwanda. [17]These authors contributed equally: Yvan Butera, Enatha Mukantwari, Maria Artesi, Jeanne D'Arc Umuringa, Vincent Bours, Andrew Rambaut, Sabin Nsanzimana, Guy Baele, Keith Durkin, Leon Mutesa, and Nadine Rujeni. ✉email: guy.baele@kuleuven.be; kdurkin@uliege.be; lmutesa@gmail.com; nrujeni@gmail.com

The coronavirus disease 2019 (COVID-19), due to severe acute respiratory syndrome coronavirus 2 (SARS-CoV-2), continues to impose a heavy death toll globally and represents a major global health challenge. The SARS-CoV-2 is a single-stranded positive-sense ribonucleic acid (RNA) virus that typically undergoes one to two single nucleotide mutations per month. Real-time whole-genome sequencing provides invaluable insights on the pandemic's transmission dynamics and enables effective surveillance. Moreover, genomic data provide useful information required for the ongoing development of vaccines, therapeutics, and diagnostic tools. Analysis of SARS-CoV-2 mutations is particularly crucial when these affect epitopes involved in the induction of host immune responses as they may lead to immune evasion, with potential implications for vaccine (and immunotherapy) efficacy.

The global SARS-CoV-2 lineage nomenclature has already been proposed with A and B as the initial epidemiological lineages representing the two original haplotypes in Wuhan[1], followed by a number of sub-lineages. As described by Rambaut et al. ref. [1], Pango lineages are monophyletic clusters of SARS-CoV-2 that are linked to an epidemiological event. Such an event can be an introduction into a distinct geographic area, evidence of increased transmission or a series of functionally relevant mutations. Variants of SARS-CoV-2 are defined by having a constellation of biologically relevant mutations, and many variants are now being monitored closely by the WHO and other public health agencies around the world. Variants may correspond to lineages directly as they operate on the same resolution, but some variants do not (e.g. B.1.1.7 + E484K is a variant, but does not correspond to a specific lineage as it has arisen many times independently). A number of variants of concern (VOCs) have been formalized by the WHO such as the Alpha VOC (B.1.1.7, 20I/501Y.V1 or VOC 202012/01), characterized by 23 mutations (13 non-synonymous mutations, four deletions and six synonymous mutations), that is associated with higher transmissibility[2] and increased mortality;[3,4] and the Beta VOC (B.1.351 or 20H/501Y.V2) that emerged independently of B.1.1.7, shares some mutations with the B.1.1.7 VOC and has recently also been associated with low vaccine efficacy in South Africa[5]. In Rwanda, the first case of SARS-CoV-2 was confirmed in the capital city of Kigali on March 14th 2020, following a series of testing at the borders and the Kigali International Airport, (KIA), and was linked to incoming travelers from Mumbai, India. Subsequently, a countrywide total lockdown, coupled with strict prevention measures including contact tracing, was enforced for nearly 2 months aiming to contain the spread of the virus (Fig. 1A and Supplementary Table S1). From May 2020, lockdown restrictions were lifted progressively, a number of commercial activities resumed and the KIA reopened on the 1st of August 2020 (Supplementary Table S1). However, despite continued massive testing[6], contact tracing, hotspot mapping, and preventive measures[7], the number of cases continued to increase (Fig. 1 and Supplementary Figure S1), mainly associated with cross-border land travels through truck drivers[8] and imported cases (Supplementary Fig. S2). This culminated in a 'first wave' of local transmission between July and September 2020. Additional containment measures led to the decline of cases until November 2020 when schools and most activities resumed. In December 2020, another 'wave' of infections hit the country, peaking in January–February 2021. As a result, new movement restrictions were enforced, including a total lockdown in the capital city and a 7 days' quarantine for international travelers in addition to two negative polymerase chain reaction (PCR) tests, one pre-departure and another one upon arrival (Supplementary table S1).

In this study, we reconstruct the introduction and subsequent dispersal of lineages A.23.1 and B.1.380 based on genomic analysis of isolates from the first and second waves of the epidemic in Rwanda. In particular, we highlight a shift from the ancestral dominant B.1.380 lineage in the early stages of local transmission to a new lineage, A.23.1, that is currently dominating throughout the country. Combining the collected genomic sequence data with associated individual travel histories to perform travel history-aware phylogeographic inference, we infer introductions into Rwanda from all of its surrounding countries including those from which no genomic sequences are available. Given the importance of these findings on regional surveillance of SARS-CoV-2, we emphasize the need for strengthening genomic surveillance at the country's points of entry following the detection of the first cases of the B.1.1.7 and B.1.351 VOCs among travelers arriving at KIA.

## Results

**Patient characteristics**. As of the 10th February 2021, a total of 16,865 cases have been confirmed in the country and the sequences analyzed represent 1.2% of the total confirmed cases. The proportion of daily confirmed cases versus the number of sequences taken is illustrated in Fig. 1. In Supplementary Fig. S1, we show a breakdown per month of these numbers, illustrating differences in genome sampling intensity compared to case counts throughout our study period, with the difference being most pronounced during the first 6 weeks of 2021. We sampled a total of 203 cases (reflecting the national screening efforts at points of entry and emerging hotspots) with an average age of 36.7 years, of whom 131 were males and 70 females (and two unknowns) in this study. Of these, location data were available for 152 individuals, of whom 99 lived in Kigali while others were living in different districts of the country. Significant efforts were made to obtain associated metadata for all cases, with specific attention to individual travel history data, as these may shed light on the origins of viral variants introduced from neighboring countries (Supplementary Data 1). Of the 203 cases, 28 had recorded travel history (mainly sampled at the airport and other points of entry through national monitoring and testing efforts) from Tanzania (6), Kenya (4), Demographic Republic of Congo (3), Uganda (3), United States of America (2), United Arab Emirates (2), South Sudan (1), Italy (1), Morocco (1), Senegal (1), Canada (1), China (1), Gabon (1), and Burundi (1). We show the origin of these collected travel cases for which we have genomic data in Fig. 2, with a focus on neighboring countries, which reveals that most travel cases originated in Tanzania, a country that has not yet made any genomic data available. Other important countries from which travelers originated were Kenya, Uganda, and Burundi, representative of the collected data on infected travelers arriving in Rwanda via air travel (Supplementary Fig. S2). We show the number of genome sequences and individual travel cases into Rwanda from these countries in Fig. 3. We also provide the GISAID accession identifiers associated with these genomes in Supplementary Table S2. For many African countries, limited to no sequences were available in GISAID, with sequencing heterogeneously clustered throughout the time period considered in this study. The availability of travel history data is thus critical in these cases as it allows us to characterize the viral population in these countries despite the absence of samples. We note that the travel cases from surrounding countries originate from the second half of 2020, with no such data being available from earlier time periods.

**Lineage characterization**. The available genomes were analyzed using the Pangolin module[1]. We show in Fig. 3 that the majority

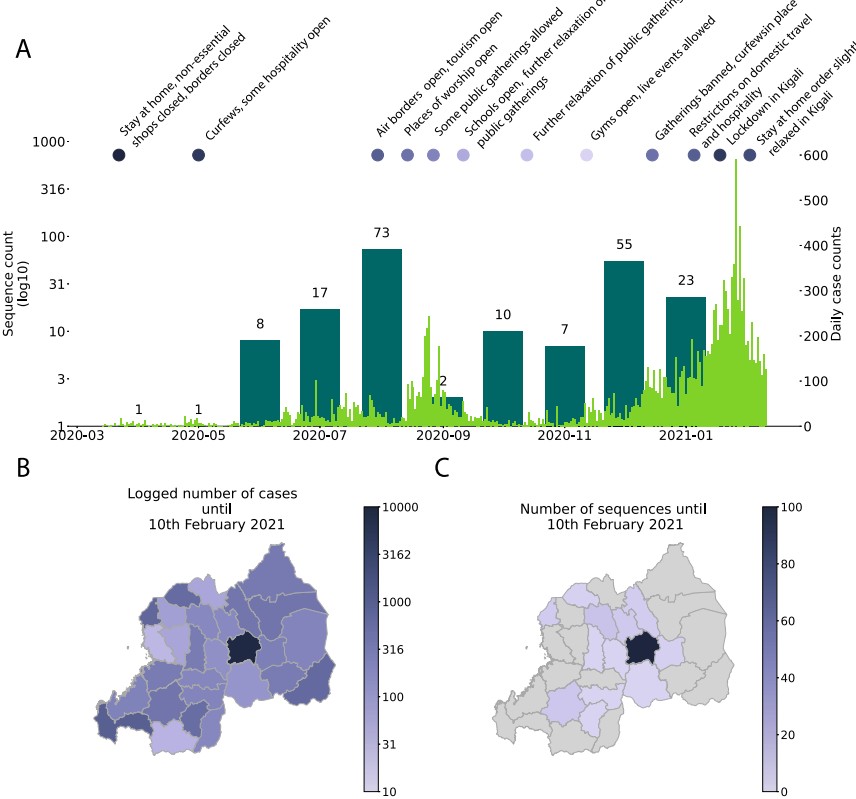

**Fig. 1 Comparison of the number of sequences taken and case counts over time and space. A** time series of month of sequence collection date (thicker bars), with thinner bars the daily new cases reported nationally until the 10th of February, 2021. Control measures are shown above the time series as colored dots, with darker colors representing more restrictions; **B** shows the number of sequences in this study by the district of residence; **C** log-transformed number of cumulative cases by the district until the 10th of February, 2021.

of the SARS-CoV-2 sequences in Rwanda belong to two distinct lineages, A.23.1 and B.1.380. However, the dynamics of their distribution changed over time, as shown in Fig. 4. Indeed, the early stages of local transmission were characterized by circulation of a dominant B.1.380 lineage, which has only been observed in Rwanda and Uganda. The diversity of the viral strains observed in the period of May to July 2020 are most likely early imports from Europe and Asia before suppressive measures (such as the countrywide lockdown and the airport closure; see Supplementary Table S1) were enforced. Nevertheless, an increased strain diversity is observed from the period August–October 2020, most likely reflecting introductions through cross-border land travels for goods and cargo[8].

Towards the end of 2020, we observed a selective sweep, with lineage A.23.1 taking over. This sub-lineage, first observed in Uganda in late 2020 was reported to contain at least four amino acid changes in the spike protein and amino acid changes in the proteins nsp3, nsp6, ORF8, and ORF9[9]. In particular, these authors suggest that the Q613H mutation in spike may be functionally equivalent to the D614G mutation that arose early in 2020 and is associated with increased viral transmissibility[10]. Bugembe et al. ref. [9]. describe a selective sweep across Uganda of this lineage, which is now the dominant lineage circulating in Uganda as well. Rwandan genome sequencing shows the presence of A.23.1 as early as October 21st 2020 and a sweep of this lineage was observed from late November (Fig. 4). A.23.1 continues to be the dominant lineage within Rwanda up until February 2021. More recently a number of infections associated with travel have been identified as variants of concern. The first import cases of B.1.1.7 and B.1.351 variants were sampled on December 28th 2020 and January 4th 2021, respectively. Analysis by Volz et al.

ref. [2]. suggests that B.1.1.7 is a more transmissible lineage, with a recent study suggesting that B.1.1.7 is not only more transmissible than preexisting SARS-CoV-2 variants, but that it may also cause more severe illness and is associated with increased mortality[4]. However, data inclusive of this paper do not report onward transmission of these VOCs.

**Phylogeographic reconstruction accommodating individual travel histories.** We made use of publicly available data and the sequenced Rwandan SARS-CoV-2 genomes - all available in GISAID[11,12] (Supplementary Data 2) - to infer a time-scaled phylogenetic tree using maximum-likelihood inference (see "Methods"). This phylogeny enabled us to identify two subtrees with predominantly Rwandan sequences (Supplementary Fig. S3). Both of these subtrees consist of genetically distinct lineages, with the larger cluster belonging to lineage B.1.380 (and hence referred to as subtree B.1) and the smaller one to A.23.1 (referred to as subtree A). The considerable difference in sampling dates and genetic distance between the sequences suggests that the currently circulating SARS-CoV-2 Rwandan lineages are a result of at least two independent introduction events that established local transmission. Subtrees A and B.1 have 172 and 218 sequences, and contain a total of 49 and 134 Rwandan sequences, respectively.

To more accurately understand the pattern of SARS-CoV-2 introductions into Rwanda, we performed a Bayesian discrete phylogeographic analysis on subtrees A and B.1 (Supplementary Data 3, 4). The 172 genomes in subtree A originated from 33 locations and included all sequences from lineage A.23.1. The 218 genomes in subtree B originated from 37 locations and included the B.1.380 lineage. In our analysis of both subtrees, we fit a travel

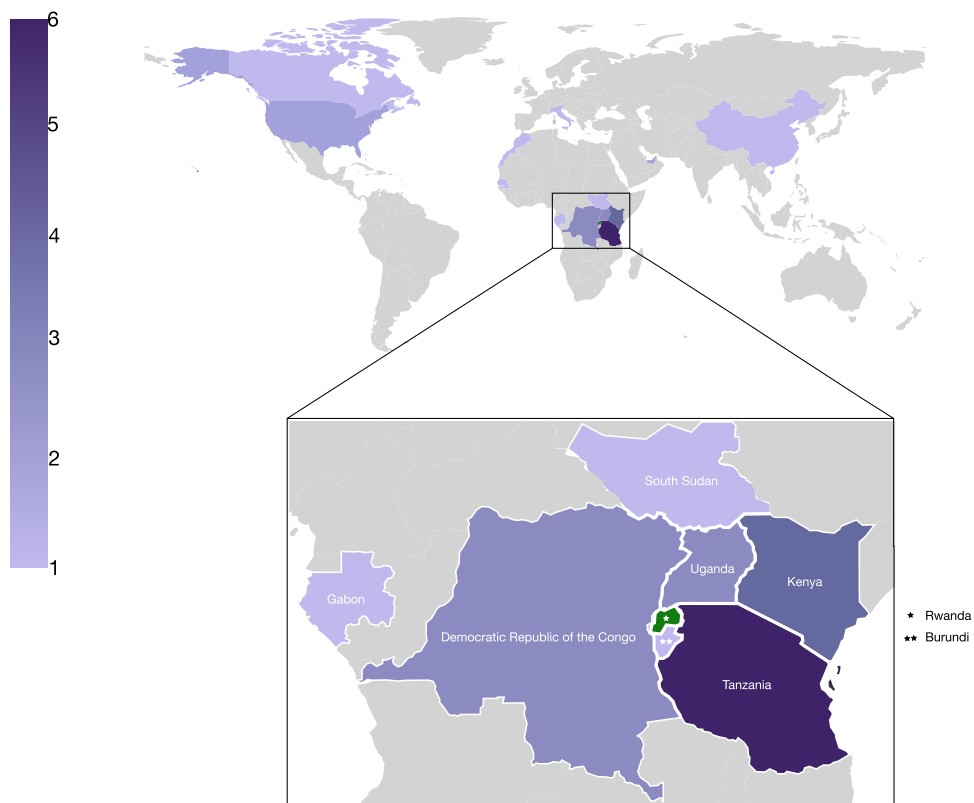

**Fig. 2 Map showing the number of sequences with recorded travel history per country ($n = 28$).** While there is travel into Rwanda recorded from across the world, most cases are from neighboring countries, notably Tanzania (6), Kenya (3), Demographic Republic of Congo (3), Uganda (3), South Sudan (1), Gabon (1), and Burundi (1).

history-aware asymmetric discrete-state diffusion process to model the spatial spread between countries. Our phylogeographic reconstructions included a total of 17 sequences with travel history, 11 for the analysis of subtree A and six for subtree B.1. Interestingly, some of these sequences have associated travel histories originating from Tanzania (four in subtree A and one in subtree B.1), a country that had not reported any COVID-19 cases since May 8th, 2020[13], and also has no publicly available genomes on GISAID. While Burundi and South Sudan have been consistently reporting case numbers, no genomic sequences are available on GISAID from these countries yet. Our joint phylogeographic reconstructions are able to include those countries as locations with SARS-CoV-2 infections (which can then be considered as possible ancestral locations), by exploiting data on infected incoming travelers from those countries. This type of phylogeographic reconstruction enables to more accurately reconstruct the spread of pathogens by exploiting additional observed data, in the form of documented individual travel histories (which don't need to be inferred).

Figures 5 and 6 show the estimated location-annotated phylogenies that enable to track the geographic spread of SARS-CoV-2 through time for subtrees A and B.1, with a focus on the available Rwandan sequences. In our analysis of subtree A (Fig. 5), which contains sequences from lineages A.23 and A.23.1, we inferred a minimum number of 22 (HPD 95%: [16–29]) introduction events into Rwanda, with a minimum of respectively 13 and 4 of these events originating from Uganda and Kenya (Fig. 7 and Supplementary Table S3). We found an expected number of two introduction events from Tanzania into Rwanda, corresponding to and being derived from the two arriving traveler cases, as well as single introduction events from South Sudan and China into Rwanda. Figure 5 also shows frequent mixing between

Rwanda, Uganda, and Kenya, with the latter two estimated to have seeded introductions into Tanzania, from where no genomic sequences are available to date. However, by carefully collecting metadata of infected individuals, we are able to confirm the presence of lineage A.23.1 among travelers from Tanzania, despite the absence of genomic data. Our travel history-aware inference methodology further enables us to consider such unsampled countries to determine the intensity of exchanges between countries and potentially even infer one of the unsampled countries as the origin of the lineage. In our analysis of subtree B.1, which includes Rwandan lineage B.1.380, we inferred a minimum number of nine (HPD 95%: [8–12]) introduction events into Rwanda, with three of these events originating from Kenya (Fig. 7 and Supplementary Table S3). We also found an expected number of two introduction events from both Uganda and Italy. Using Bayesian stochastic search variable selection (BSSVS), we identified seven statistically supported (Bayes Factor > 3) transition routes into Rwanda for subtree A and six for subtree B.1 (Supplementary Table S3). Our analysis on subtree A showed that Uganda accounted for the majority of SARS-CoV-2 introductions into Rwanda (mean number of Markov jumps: 13.1; 95% HPD: [7–20]), whereas our analysis on subtree B.1 identified Kenya as the main source of SARS-CoV-2 introductions into Rwanda (mean number of Markov jumps: 3.2; 95% HPD: [0–5]).

Consistent with previously published analyses of SARS-CoV-2, we observe that our discrete Bayesian phylogeographic reconstructions resulted in MCC trees of which the internal nodes can be poorly supported, a common phenomenon in SARS-CoV-2 phylogenies (Figs. 5 and 6). The considerable uncertainty in phylogenetic clustering results in a variety of diverging phylogeographic histories, which end up not being captured in the MCC

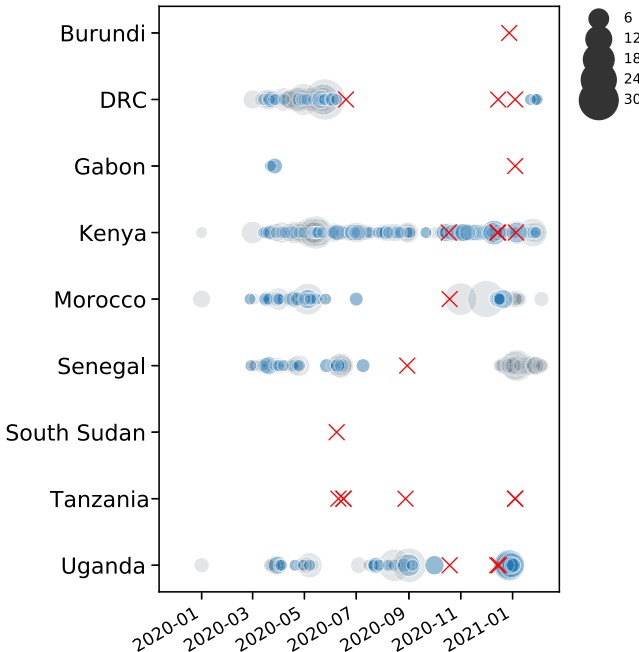

**Fig. 3 Availability of whole genome sequences for African countries from which travelers entered Rwanda.** Gray circles denote the number of sequences available in GISAID for each country on a given date. Blue circles correspond to the number of sequences included in our analyses. Red crosses mark the collection dates of Rwandan sequences with travel history from the respective countries. Although few to no sequences are available from Burundi, Gabon, South Sudan, and Tanzania, these travel history data point to SARS-CoV-2 lineages circulating in these countries, to the extent that returning travelers from these countries import those lineages into Rwanda.

trees as these only represent point estimates of the posterior distribution. To this end, we explored the ancestral spatial histories of individual samples of interest using Markov jump trajectory plots[14,15] (Fig. 8). In the case of subtree A, the travel-aware reconstructions showed four sequences consistently forming two clusters with posterior support > 0.9. However, the first two of these four cases correspond to cross-border truck drivers of Tanzanian nationality (sampled on the same day on the same sampling location, i.e. the Rusumo border), with no such metadata available for the other two cases in subtree A. Hence, the two inferred introductions actually correspond to four introduction events from Tanzania into Rwanda, which are clustered together by location in our joint inference, likely as a result of additional samples currently lacking from the border region. Because of this, sequences in each cluster result in nearly identical spatial histories. Fig. 8A, B show the Markov jump trajectory plots for these two introductions. Overall, we see considerable ambiguity in the ancestral locations prior to Tanzania, as seen by the density of lines landing in "Other" alternate locations. More broadly, we see that in both cases the Rwandan sequences diverged from ancestors in Tanzania, Kenya, and Uganda, with considerable uncertainty placed at the root, among the Democratic Republic of the Congo (DRC), Sierra Leone, and Mali. The introduction in subtree B, on the other hand, presents us with a different ancestral relation with Tanzania (Fig. 8C). Although we also generally observe considerable uncertainty in the ancestral paths, we observe a strong signal for an ancestry in Rwanda prior to the introduction from Tanzania. This would imply a transmission chain starting in Rwanda, spreading into Tanzania, and then being reintroduced into Rwanda. A similar dynamic of outflow and inflow of Rwandan lineages can be seen in the ancestral histories for the sequences with travel history to Morocco, Italy, and the DRC (Supplementary Figure S5). This suggests a bidirectional exchange of SARS-CoV-2 genomes between each of these

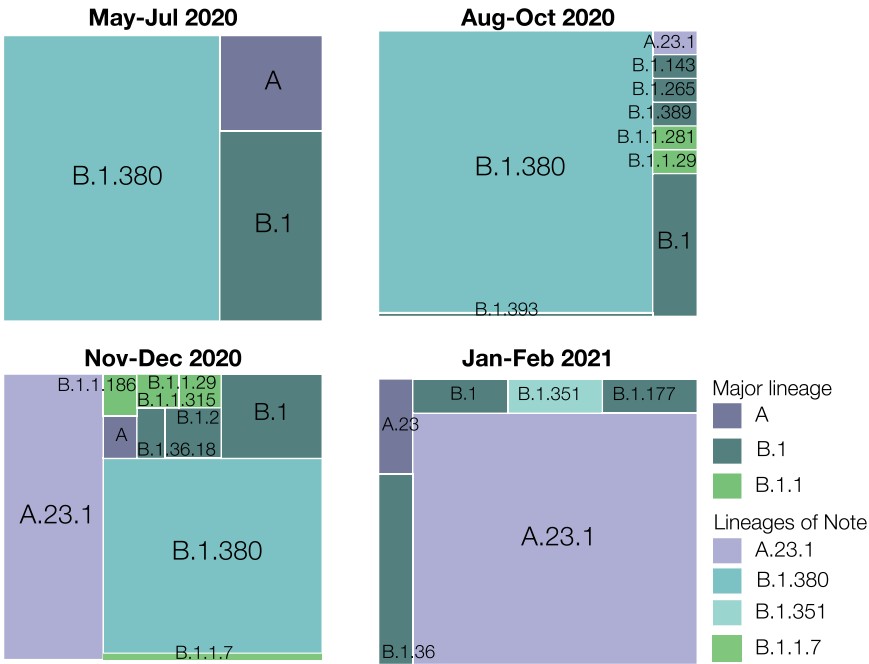

**Fig. 4 Lineage diversity sampled in Rwanda across four time points: May–Jul 2020 ($n = 28$), Aug–Oct 2020 ($n = 86$), Nov–Dec 2020 ($n = 74$), Jan–Feb 2021 ($n = 28$).** Lineage B.1.380, a Rwanda-specific lineage, dominated the sampled diversity during the first wave. Lineage A.23.1 first appeared in Rwanda in October 2020, and quickly attained a significant proportion of the sampled SARS-CoV-2 genome sequences. More recently, we detected and sequenced single cases of the B.1.1.7 and B.1.351 VOCs, associated with incoming travelers from Burundi and the Democratic Republic of the Congo, respectively. $n$: number of positive SARS-CoV-2 samples sequenced.

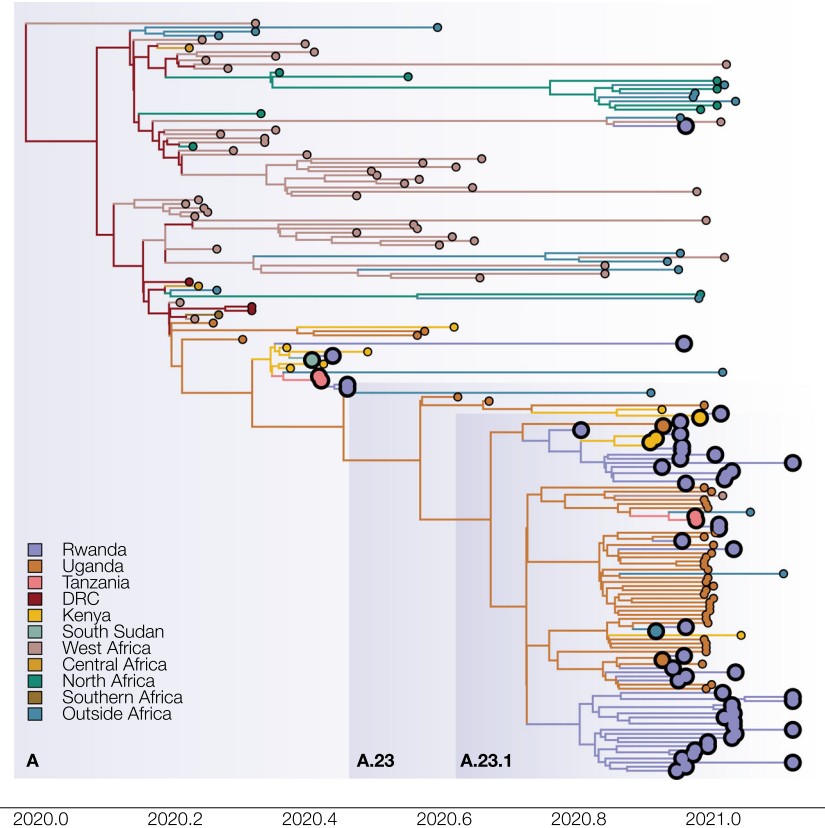

**Fig. 5 Maximum clade credibility phylogeny for subtree A, representing diversity of lineages A.23 and A.23.1.** The phylogeny with associated ancestral locations was inferred using travel history-aware asymmetric discrete state phylogeographic inference. A total of 33 locations were considered in the analysis but are grouped for visualization purposes. The branches in the phylogeny are colored according to the geographical location of the reconstructed ancestral regions. Rwandan sequences are indicated as large tips, colored by associated travel histories (available for 11 of the Rwandan sequences). The travel history-aware phylogeographic reconstruction on subtree A infers frequent mixing between Rwanda, Uganda, and Kenya, with the latter seeing introduction events from both Uganda and Rwanda. Both Kenya and Uganda are estimated to have seeded introductions into Tanzania, with the former also seeding an introduction into South Sudan. Importantly, the travel history-aware approach includes (returning infections from) Tanzania in lineage A.23.1, which could not be inferred via other phylogeographic approaches.

countries and Rwanda. However, because of the differences in sequencing efforts across the globe, we cannot dismiss the possibility of intermediary locations in these cases. Nonetheless, all spatial trajectory plots imply the presence of SARS-CoV-2 lineages circulating in Tanzania after May 2020. The difference in ancestral histories coupled with the fact that these travel history sequences are genetically distant from each other implies that multiple SARS-CoV-2 lineages have circulated in Tanzania to this day.

In addition, subtree A contains a sequence with travel history to South Sudan. Although over 10,000 COVID-19 cases have been reported to date[13], no genomic sequences were publicly available for South Sudan before June 2021. The sample tested at arrival in Rwanda presents us with evidence of lineage A having circulated in South Sudan during the months of May and June 2020 (Fig. 8D). As expected, the Markov jump trajectory plots for this sample also show considerable uncertainty in the reconstruction of the ancestral locations prior to South Sudan. Regardless, we see some support for ancestry in Kenya, Uganda and the DRC, which provides further evidence for viral transmission between the neighboring countries in the area. We compare in "Supplementary Materials" the diversity of lineages in Rwanda to that of two of its neighboring countries that have released a similar number of SARS-CoV-2 genomes, i.e. Uganda and Kenya.

In Supplementary Fig. S6-S8, we show that, while each country has its own dynamics, Rwanda and Uganda have seen a similar rise in the number of infections with lineage A.23.1, whereas the surge in infections with B.1.380 was specific to Rwanda. In Supplementary Fig. S9 and S10, we show the differences between the number of recorded travel cases and the estimated Markov jumps in both subtrees A and B.1, illustrative of the ability of travel history-aware phylogeographic reconstruction to estimate transition between countries beyond what has been collected as part of the metadata associated with the available genomes.

To assess whether virus populations were structured per country, we performed compartmentalization analyses using tree-based methods on a posterior distribution of phylogenies in BaTS[16]. BaTS yielded significant values for all three statistics: $p < 0.001$ for PS and AI, $p < 0.01$ for MC(Rwanda) in both of the A.23.1 and B.1.380 subtree analyses. The significant degree of clustering suggests that for both of these lineages, local transmission chains have played an important role in driving the Rwandan epidemic. Because we find a significant tendency for Rwanda SARS-CoV-2 genomes to cluster according to the location of sampling, we subsequently investigated the spatial patterns of virus spread within Rwanda. Our continuous phylogeographic analysis of SARS-CoV-2 lineages highlight an important inter-connection of those lineages centered around

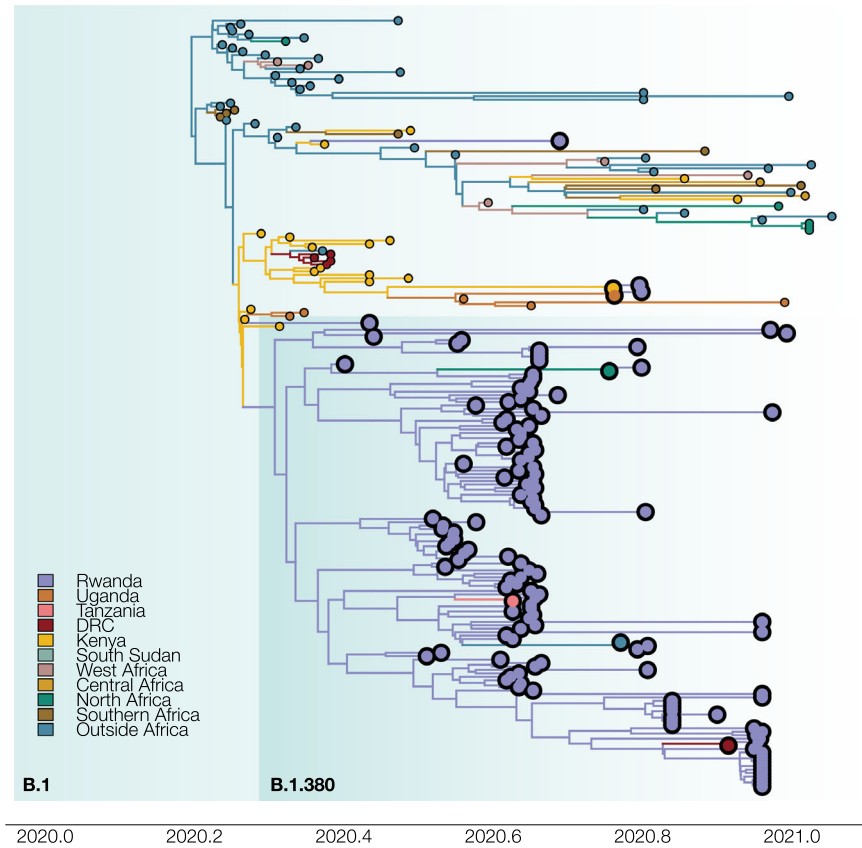

**Fig. 6 Maximum clade credibility tree for subtree B.1, which includes Rwandan lineage B.1.380.** The phylogeny with associated ancestral locations was inferred using travel history aware asymmetric discrete state phylogeographic inference. A total of 37 locations were considered in the analysis. The branches in the phylogeny are colored according to the geographical location of the reconstructed ancestral regions. Rwandan sequences are indicated as large tips, colo red by their associated travel histories. A total of six Rwandan sequences with associated travel history are highlighted in this subtree. The travel history-aware phylogeographic reconstruction on subtree B.1 infers a large local transmission cluster in Rwanda (subtree B.1.380). However, by incorporating individual travel histories into the phylogeographic reconstruction, we are able to infer that this subtree does not solely represent local transmission, but also introduction events into Rwanda from Tanzania, Morocco, South Sudan, and the Democratic Republic of the Congo.

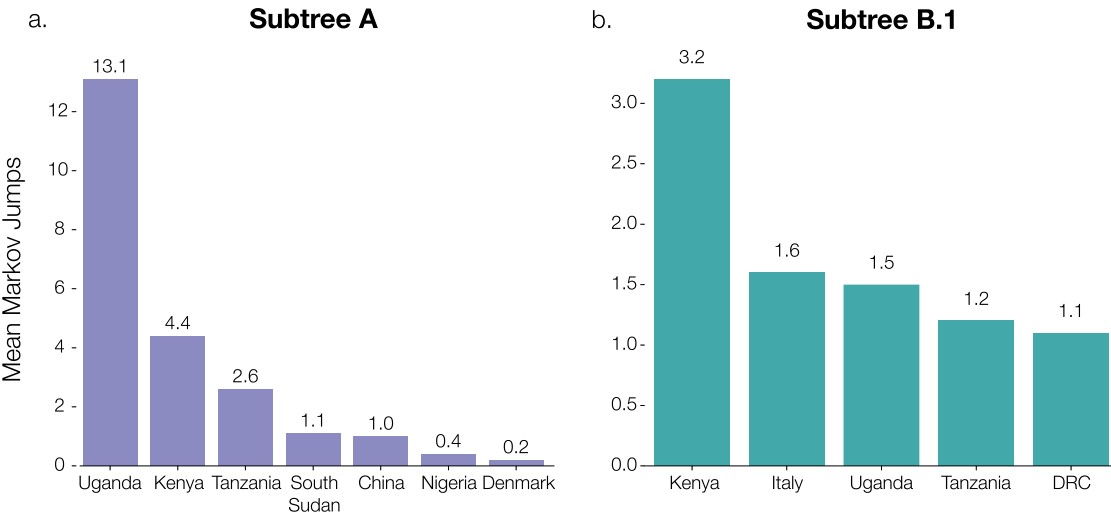

**Fig. 7 Supported transitions into Rwanda.** Mean number of Markov jumps for supported transition rates into Rwanda (Bayes Factor > 3) for subtrees A and B.1. Support for these rates was determined using BSSVS with a travel history-aware asymmetric discrete phylogeographic model on both subtrees A and B.1. In both analyses, the majority of introductions into Rwanda were inferred to originate from nearby countries in East Africa, suggesting a substantial exchange of viral lineages between neighboring countries in the region. We refer to Supplementary Table S3 for the Bayes factor support values for these reported transitions.

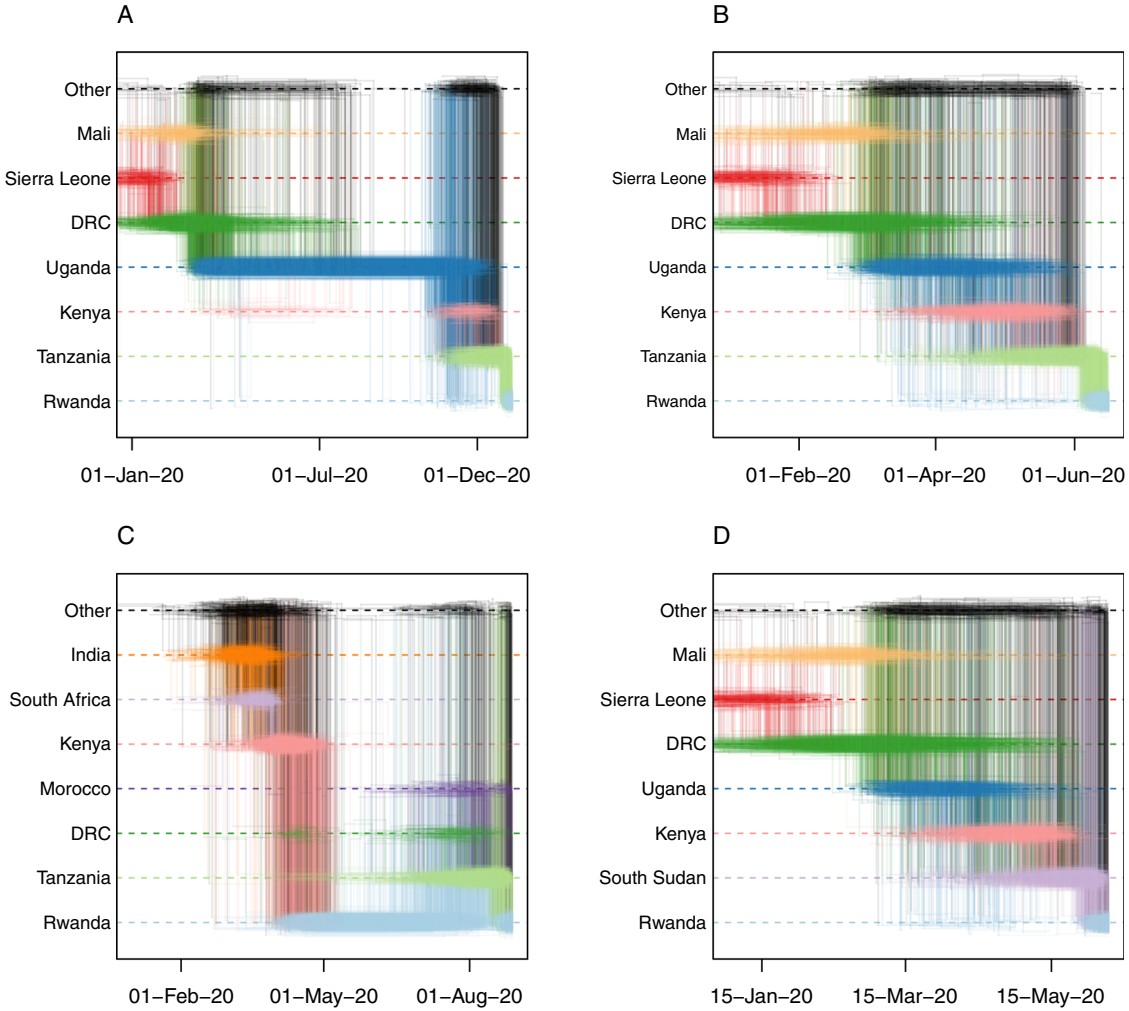

**Fig. 8 Ancestral spatial trajectories for individual patients.** Markov jump trajectory plots for four selected Rwandan infected individuals with travel history (returning) from Tanzania (A, B, C) and South Sudan (D). Each individual trajectory corresponds to the Markov jumps in a single tree from the posterior distribution, with each plot showing the uncertainty across a subsample of 1,000 posterior trees. The horizontal dimension represents the time maintained at an ancestral location. Vertical lines represent a Markov jump between two locations. The seven most prominent locations across all ancestral paths in the posterior are displayed along the Y-axis, with "Other" representing the remaining locations. Trajectory plots **A**, **B**, and **D** correspond to the isolates in subtree A, i.e. EPI_ISL_1064164, EPI_ISL_707772, and EPI_ISL_707712, respectively. Trajectory plot **C** corresponds to isolate EPI_ISL_960250 in subtree B.1. In all cases, considerable uncertainty in the ancestral reconstructions can be seen from the pattern of overlapping horizontal lines and the diffuse density of vertical lines, which indicate considerable support for different ancestral locations (i.e. uncertainty in the spatial reconstruction), and variance in the reconstructed timing of the introductions. For trajectory plots **A**, **B**, and **D**, we observe similar patterns in the spatial paths reconstructed, where the isolates find ancestries in Mali/Sierra Leone/Democratic Republic of Congo, Uganda, and Kenya prior to each corresponding travel location. In contrast, trajectory plot **C** shows support for an ancestry in Rwanda prior to the virus circulating in Tanzania and being reintroduced into Rwanda. This indicates a bidirectional exchange of viral lineages between the two countries, although the possibility of an intermediary country being involved cannot be discarded due to unevenness in sampling efforts between countries.

Kigali, after having been introduced in Rwanda (Supplementary Fig. S11). Most sequences sampled outside the city appeared to be evolutionarily linked to sequences sampled within this city area, and would then correspond to independent dispersal events from Kigali. However, this phylogeographic pattern, i.e. the central importance of Kigali within the dispersal history of SARS-CoV-2 lineages, might to some extent result from the higher sampling effort within the capital city. To assess the effect of sampling bias, we performed multiple replicate analyses on subsampled data sets, showing a consistent pattern of spread with the one inferred on the original data set (Supplementary Fig. S12; see "Supplementary Materials" for additional information on the sensitivity analysis performed). Regardless, it is likely that a higher sampling effort outside Kigali would highlight more local transmission and that this represents an important avenue of further research.

## Discussion

Here we describe the pattern of transmission of SARS-CoV-2 in Rwanda from May 2020 to February 2021. In particular, we report the spread of a SARS-CoV-2 variant of the A lineage (A.23.1) with notable amino acid changes in the spike protein as well as several non-spike protein changes first detected in Uganda[9]. Indeed, most SARS-CoV-2 sequence diversity in Rwandan strains belong to two distinct lineages: A.23.1 and B.1.380. The latter dominated throughout the early stages of the pandemic before a shift towards the A.23.1 lineage occurred in November 2020. A similar pattern was observed in neighboring Uganda as described by Bugembe et al. ref. [9]. The authors describe the lineage as a variant of concern (VOC) in a sense that it shares mutations with the currently known lineage B VOCs such as the changes in key spike protein regions (the furin

cleavage site and the 613/614 change). However, functional analyses are needed to determine whether these mutations have effects on transmission rates, immune evasion, vaccine efficacy, and/or case-fatality rates.

In this study, we reported on the ongoing genomic sequencing efforts in Rwanda, which are complemented with careful collection of associated travel history metadata of incoming travelers. These efforts enabled us to exploit this information by performing joint Bayesian travel history-aware phylogeographic inference on these data. By applying this recently developed approach, we demonstrated considerable contributions of neighboring countries' sequence introductions into Rwanda (as well as possible bidirectional exchanges). Of particular interest to this study, we were able to include traveler cases from Tanzania, Burundi, and South Sudan while none of these three countries had made any SARS-CoV-2 genomes available throughout the study period. According to the data we collected, two infected Rwandan travelers returned from Tanzania on the 16[th] of June, 2020 and two more on 4[th] January, 2021. Our findings also complement a statement from the WHO[17] that a number of travelers from Tanzania who have traveled to neighboring countries and beyond have tested positive for COVID-19. Incorporating travel history information in phylogeographic analysis can mitigate sampling bias (from unsampled or under-sampled countries)[14], although this cannot fully replace the lack of sequences from other countries.

The reported import into Rwanda of two VOCs, i.e. B.1.1.7 and B.1.351, sampled at the Kigali International Airport in late December 2020 and early January 2021 are of particular interest. The patient infected with the B.1.1.7 variant was a Burundian traveling from Burundi while the patient infected with the B.1.351 variant was a Zimbabwean coming from the DRC, suggesting that VOCs may be actively circulating in neighboring countries. Indeed, although Burundi and Tanzania have currently no SARS-CoV-2 sequences uploaded onto GISAID, and South Sudan not until June 2021, the DRC has shared a total of 416 sequences, of which 21 are VOCs (eight B.1.1.7 and 13 B.1.351), while Kenya has shared a total of 1478 sequences.

Ongoing genomic surveillance in Rwanda revealed additional infections with these VOCs (mostly B.1.351) from travelers sampled at the airport. In an effort to curb the spread of the different lineages and variants, and following the upsurge of cases in November–December 2020, several measures were taken by the Rwandan government including a 7-day quarantine to all incoming passengers followed by an RT-PCR test, in addition to presenting a COVID-19 negative test upon arrival. Furthermore, the capital city of Kigali went through a total lockdown from mid-January to early February 2021, and travels between districts were prohibited until mid-March 2021. A 7 pm to 4 am curfew was instituted in early February 2021; public offices were closed and employees were working from their homes. All schools in Kigali were closed as well, and classes were being held online. Cafés and restaurants were only providing takeaway services. Churches, public swimming pools, and gyms were closed (Office of the Prime Minister - Republic of Rwanda 2021; Supplementary Table S1). Such suppression mechanisms (population-wide social distancing, lockdown, school closure, case isolation) have been shown to have the greatest impact (as far as non-pharmaceutical approaches are concerned) in terms of transmission control[18]. Additionally, all public health facilities received free antigen rapid diagnostic tests for every single person presenting COVID-19 related symptoms. Moreover, a vaccination campaign was initiated in March 2021, with the aim to vaccinate all front liners and vulnerable populations (elderly and people with other underlying health conditions) in the first phase. To this end, Rwanda received both Pfizer and AstraZeneca vaccines. A rapid and efficacious vaccination coverage will ease the social and economic disruptions associated with non-pharmaceutical interventions. However, a number of published studies[19–21] demonstrate evidence of escape of SARS-CoV-2 VOCs from vaccine-induced immunity. For example, Becker et al. ref. [21]. reported a 'substantially reduced Ab neutralization for the B.1.351 variant' on sera obtained from vaccinated people, highlighting the importance of genomic surveillance, monitoring incoming travelers, and efficient contact tracing upon appearance of new variants.

Our results suggest that neighboring countries play an important role in establishing the circulation of (different strains of) SARS-CoV-2 in Rwanda. However, due to the unevenness in sampling across countries, with several not yet having provided any genomic sequences, additional data are required to accurately assess the effect of short-distance (e.g. crossing the borders with neighboring countries) versus long-distance travel in shaping the Rwandan epidemic.

## Methods

**Study design**. This is an in-depth study of SARS-CoV-2 strains that circulate in Rwanda from May 2020 to February 2021, in which we describe the demography and epidemiology of 203 SARS-CoV-2 genomes from collected SARS-CoV-2 positive oropharyngeal swabs. These swabs were obtained from two distinct groups: from individuals residing in different provinces of Rwanda ($n = 189$) and from returning travelers, whose samples were collected at the airport ($n = 14$). All samples were extracted from the biorepository of the National Reference Laboratory (NRL), in Kigali, Rwanda. Samples with a cycle threshold ($Ct$) value below 33 were selected, ensuring a wide geographical representation as well as ports of entry, and case description variables (date and place of RT-PCR test, age, gender, occupation, residence, nationality, travel history) were reported.

### Sequencing

*RNA Extraction*. Ribonucleic acid (RNA) of the virus was extracted from confirmed SARS-CoV-2 positive clinical samples with Ct values ranging from 13.4 to 32.7 on a Maxwell 48 device using the Maxwell RSC viral RNA kit (Promega) following a viral inactivation step using proteinase K according to the manufacturer's instructions.

**SARS-CoV-2 whole genome sequencing**. Reverse transcription was performed using SuperScript IV VILO master mix, and 3.3 µl of RNA was combined with 1.2 µl of master mix and 1.5 µl of H₂O. This was incubated at 25 °C for 10 min, 50 °C for 10 min, and 85 °C for 5 min. PCRs used the primers and conditions recommended in the nCoV-2019 sequencing protocol (ARTIC Network, 2020) or the 1,200 bp amplicons described by Freed and colleagues[22] (Supplementary Table S4).

Primers from version 3 of the ARTIC Network and the 1,200 bp amplicons were used and were synthesized by Integrated DNA Technologies. Samples were multiplexed using the Oxford Nanopore native barcoding expansion kits 1–12, 13–24, or the native barcoding expansion 96 in combination with the ligation sequencing kit 109 (Oxford Nanopore). Sequencing was carried out on a MinION using R9.4.1 flow cells.

**Genome assembly**. The data generated via the Oxford Nanopore Technology (ONT) MinION was processed using the ARTIC bioinformatic protocol (https://artic.network/ncov-2019/ncov2019-bioinformatics-sop.html). Briefly, the FAST5 sequence files were base called and demultiplexed using Guppy 4.2.2 in high accuracy mode, requiring barcodes at both ends of the read. FASTQ reads associated with each sample were filtered and concatenated via the guppyplex module. Consensus SARS-CoV-2 sequences were generated via the ARTIC nanopolish pipeline and assembled for each sample by aligning the respective sample reads to the Wuhan-Hu-1 reference genome (GenBank Accession: MN908947.3) with the removal of sequencing primers, followed by a polishing step using the raw Fast5 signal files. Positions with insufficient genome coverage were masked with N.

**Phylogenetic and phylogeographic analysis**. We downloaded all SARS-CoV-2 genomes from the available nextstrain build[23] with Africa-focused subsampling (https://nextstrain.org/ncov/africa) on February 23, 2021. These sequences were further complemented to include all 203 Rwandan sequences generated in this study and available on GISAID on February 24, 2021. The 203 Rwandan whole-genome SARS-CoV-2 genomes were assigned Pango lineages, as described by Rambaut et al. ref. [1], using pangolin v2 and pangoLEARN model v2021-02-21 by O'Toole et al. ref. [24]. We used Squarify to construct the square treemaps of lineage diversity across three time points[25]. We mapped the combined data set against the canonical reference (GISAID ID: EPI_ISL_406801) using minimap2[26] and trimmed the data to positions 265-29,674 and padded with Ns in order to mask out 3'

and 5' UTRs. We used the resulting alignment to estimate an unrooted maximum-likelihood phylogeny (Supplementary data 5) using IQ-TREE v2.1.2[26] using its automated model selection approach that identified the general time-reversible model with empirical base frequencies and an auto-discrete-gamma model for varying rates across sites with eight rate categories (GTR + F + R8) as best fitting the data. We subsequently calibrated this phylogeny in time using TreeTime[27] while estimating the molecular clock and skyline coalescent model parameters and using three SARS-CoV-2 genomes from Wuhan, 2019, as the outgroup.

We went on to perform a discrete Bayesian phylogeographic analysis in BEAST 1.10.5[28] using a recently developed model that is able to incorporate available individual travel history information associated with the sequenced Rwandan samples[14,15]. Exploiting such information can yield more realistic reconstructions of virus spread, particularly when travelers from unsampled or under sampled locations are included to mitigate sampling bias. To this end, and given that it is not feasible to perform such an analysis on the full data set due to a large number of sequences, we selected two subtrees in the overall phylogeny (see "Results" section) that predominantly consisted of Rwandan sequences, consisting of 172 (subtree A) and 218 sequences (subtree B.1), of which, respectively, 11 and six infected individuals have associated travel history information (Supplementary Table S2). We incorporated the collection dates for those sequences into our analyses, and treated the time when a traveler started the return journey to Rwanda as a random variable given that the time of traveling to the sampling location (in Rwanda) was not known (with sufficient precision). We specify normal prior distributions over these 17 random variables informed by an estimate of the time of infection and truncated to be positive (back-in-time) relative to sampling date. As in the work of Lemey et al. ref. [14], we use a mean of 10 days before sampling based on a mean incubation time of 5 days[29], and a constant ascertainment period of 5 days between symptom onset and testing[18], and a standard deviation of 3 days to incorporate the uncertainty on the incubation time. We retrieved the 172 and 218 sequences from the full alignment and performed joint discrete phylogeographic inference on each resulting data set using BEAST 1.10.5, employing the BEAGLE 3.2.0 high-performance computational library[30] to improve performance. For each of these phylogeographic analyses, we make use of Bayesian stochastic search variable selection (BSSVS) to simultaneously determine which migration rates are zero depending on the evidence in the data and to efficiently infer the ancestral locations, in addition to providing a Bayes factor test to identify significant non-zero migration rates[31]. We also estimated the expected number of transitions (known as Markov jumps)[32] into Rwanda from all other countries in the data set. These analyses ran for a total of 200 and 250 million iterations, respectively, with the Markov chains being sampled every 100,000th iteration, in order to reach an effective sample size (ESS) for all relevant parameters of at least 200, as determined by Tracer 1.7[33]. We used TreeAnnotator to construct maximum clade credibility (MCC) trees for each subtree.

For each subtree analysis, we assessed whether the SARS-CoV-2 lineages were structured according to country. To this end, we investigated the association between phylogeny and sampling location using Bayesian Tip-association Significance testing as implemented in the BaTS software package[16]. BaTS allows testing for a significant degree of taxon-trait clustering by evaluating three different statistics: parsimony score (PS), association index (AI), and monophyletic clade (MC) size on a posterior sample of trees. These computed statistics are then compared to a null distribution of permuted taxon-trait values, corresponding to a situation of randomly mixed locations, implying a dominant role of importations over local circulation in establishing the (local) epidemic. We performed our BaTS analyses on a sample of 1000 posterior trees and computed 100 null replicates.

To explore the local spread of SARS-CoV-2 lineages introduced in Rwanda, we also performed a continuous phylogeographic analysis following a procedure similar to one defined by Dellicour et al. ref. [34]. Specifically, we used the relaxed random walk (RRW) diffusion model[35] available in BEAST 1.10.5[28] to infer the dispersal history of Rwandan lineages along Rwandan clades identified within the two subtree-specific MCC trees that resulted from the discrete Bayesian phylogeographic inference described above. To achieve a sufficient level of spatial precision, the continuous phylogeographic analysis was only based on those sampled genomes for which the Rwandan sector of origin was known, which is the maximal level of spatial precision available for these samples. For each sampled genome associated with this level of sampling precision, which corresponds to 57% of available Rwandan genomes, we retrieved geographic coordinates from a point randomly sampled within its sector of origin. The MCMC chain was run in BEAST 1.10.5 for 30 million iterations and sampled every 10,000th iteration, its convergence/mixing properties were again assessed with Tracer[33], and an appropriate number of sampled trees was discarded as burn-in (10%). The resulting sets of plausible trees were used to obtain subtree-specific MCC summary trees using TreeAnnotator, and we then used functions available in the R package "seraphim"[36] to extract spatio-temporal information embedded within posterior trees and visualize the continuous phylogeographic reconstructions. Finally, we used the baltic Python library to visualize the phylogenies[37].

**Ethical approval**. The study was approved by the Rwanda National Ethics Committee (FWA Assurance No. 00001973 IRB 00001497 of IORG0001100/ 15April2020). An exemption from informed consent was issued based on the use of retrospective anonymous data and no medical intervention. The study was further approved by the IRB of the University of Rwanda, College of Medicine and Health Sciences (Approval notice No 325/CMHS IRB/2020).

**Reporting Summary**. Further information on research design is available in the Nature Research Reporting Summary linked to this article.

## Data availability

The reported SARS-CoV-2 genomes are available on GISAID (www.gisaid.org) under the accession numbers EPI_ISL_614763, EPI_ISL_614980, EPI_ISL_615063, EPI_ISL_615064, EPI_ISL_615067, EPI_ISL_615069, EPI_ISL_615071, EPI_ISL_615074, EPI_ISL_615075, EPI_ISL_707711-EPI_ISL_707713, EPI_ISL_707771-EPI_ISL_707774, EPI_ISL_707776, EPI_ISL_707777, EPI_ISL_707779, EPI_ISL_707780, EPI_ISL_707783, EPI_ISL_707787- EPI_ISL_707790, EPI_ISL_735436-EPI_ISL_735438, EPI_ISL_735444-EPI_ISL_735448, EPI_ISL_925847-EPI_ISL_925915, EPI_ISL_930567, EPI_ISL_930634, EPI_ISL_930853, EPI_ISL_960227-EPI_ISL_960302, EPI_ISL_1063900-EPI_ISL_1063901, EPI_ISL_1063905, EPI_ISL_1063915, EPI_ISL_1063994, EPI_ISL_1064022, EPI_ISL_1064147-EPI_ISL_1064149, EPI_ISL_1064152-EPI_ISL_1064154, EPI_ISL_1064163-EPI_ISL_1064166, EPI_ISL_1064168,EPI_ISL_1064170, EPI_ISL_1064171. We have also deposited the reads used to generate the SARS-CoV-2 genomes into the European Nucleotide Archive (ENA) under the accession number PRJEB45303.

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

## Acknowledgements

This research was commissioned by the National Institute of Health Research (NIHR) Global Health Research programme (16/136/33) using UK aid from the UK Government (funding to E.M. and N.R. through TIBA partnership) and additional funds from the Government of Rwanda through RBC/National Reference Laboratory in collaboration with the Belgian Development Agency (ENABEL) for additional genomic sequencing at the GIGA Research Institute-Liege/Belgium. The views expressed in this publication are those of the authors and not necessarily those of the NIHR, the National Institute of Health Research, the Department of Health and Social Care, or the Rwandan Government. G.B. acknowledges support from the Internal Fondsen KU Leuven/Internal Funds KU Leuven (Grant No. C14/18/094) and the Research Foundation–Flanders ("Fonds voor Wetenschappelijk Onderzoek - Vlaanderen," G0E1420N, G098321N). S.L.H. acknowledges support from the Research Foundation-Flanders ("Fonds voor Wetenschappelijk Onderzoek - Vlaanderen," G0D5117N). S.D. is supported by the Fonds National de la Recherche Scientifique (FNRS, Belgium). VH was supported by the Bi otechnology and Biological Sciences Research Council (BBSRC) [grant number BB/M010996/1]. A.O.T. is supported by the Wellcome Trust Hosts, Pathogens & Global Health Programme [grant number: grant.203783/Z/16/Z] and Fast Grants [award number: 2236]. A.R. acknowledges the support of the Wellcome Trust (Collaborators Award 206298/Z/17/Z – ARTIC network) and the European Research Council (grant agreement no. 725422 – ReservoirDOCS).

## Author contributions

Y.B. and E.M.: Study design, data collection. M.A., Y.B., B.B., E.M., J.d.U.: RT-PCR, library preparation, whole genome sequencing. K.D., Y.B., S.R.: Whole genome sequencing, sequences cleaning and assembling, sequences fast files production. Y.B., E.M., S.B., O.M., J.d.U.: RNA extraction. P.T., R.S., M.G., R.R., E.U., S.D., A.K., O.M., R.M., S.G.: Sample selection, data collection. S.D., S.L.H., V.H., G.B.: Spatial and phylogeographic analysis. J.R., D.G., J.S., W.N., J.Q., M.M.M., A.K., P.C.R., N.L., J.P.R., S.N., T.M., D.N.: Provided technical guidance and review of the paper. G.B., A.O.T.: Phylogenetic analysis. V.B., J.S., W.N., G.B., A.R., S.N., K.D., M.A., L.M., N.R.: Provided scientific and technical guidance, review of the paper. V.B., N.R., G.B., K.D., L.M.: Secured funding and coordinated the study.

## Competing interests

The authors declare no competing interests.
