## [Peer Review File · Nature Communications]

REVIEWER COMMENTS

Reviewer #1 (Remarks to the Author):

The authors look at the COVID-19 epidemic in Rwanda using one year of data. The study is certainly interesting and methods are cutting-edge. However it presents a limitation: sampling bias. Looking at figure 1, the sampling bias appears quite striking: most of the sampling si coming from one region (name the region in the figure). I would like to see from the authors a supplementary figure that shows cases per region per month of sampling and corresponding breakdown of sampling per region (like panels a & b but for each month of sampling). Moreover, there are few points during which cases were decreasing or not surpassing 200 counts and for which the authors have many sequences, but when the epidemic started accumulating more cases (jan2021 and after), sampling was scarce (23 and 5 sequences, with 5 sequences coming from a time when daily new cases were almost 600) . please add :“daily cases” to y axis in fig 1c so that the reader can easily interpret the figure without reading the legend. Wouldn't the bias in sequences from Kigali bias the analysis reported in figure S5? What the authors did to prevent such potential issue?

Please make sure to describe in more detail what figure 2 and 3 indicate, otherwise move them to supplementary. Right now, it seems like the reader has to interpret them since they are only briefly mentioned in line 135.

Shouldn't paragraph “Phylogenetic analysis” called instead “lineage characterization” ?

Lines 213-215 refer to bayes factors, not markov jumps, therefore the reference to figure 7 is not correct. Markov jumps are described afterwards and that's where the ref to figure 7 is needed. I think figure S3 should not be supplementary.

There is no reference to figure S1 in the text.

Authors need to add an GISAID author acknowledgement table in supplementary with all the strains that were used from GISAID.

Besides the above mentioned limitation, the manuscript is poorly formatted. Some figures are not discussed in the text or mentioned, and some others, that might be more relevant are found in the supplementary materials rather than in the main text.

Reviewer #2 (Remarks to the Author):

In this manuscript, the authors describe a study, where they sequenced over 200 cases of SARS-CoV-2 in Rwanda. Using these sequences, they then describe the importance of introductions that have led to these cases. Using phylogenetic reconstruction, alongside travel information of sampled individuals, they reconstruct the phylogenetic tree of sequences from Rwanda alongside other sequences from the region and around the globe to determine the spatial spread of SARS-CoV-2 in Rwanda. Additionally, they find an abundance of several different Variants of SARS-CoV-2 as well as such that are designated variants of concern. The manuscript importantly fills a gap in the current literatures of SARS-CoV-2 by analyzing its spread in a region that has been under covered elsewhere and the manuscript is overall well written. I do though think though that the manuscript would greatly profit by going beyond mostly standard phylogenetic and ancestral reconstructions and largely descriptive analyses, to be a bit more quantitative. For example, there are different variants, but did they affect

anything or does the shift in variants mainly reflect what is happening elsewhere and how are the shifts in variants not just by chance? Also, the authors show that SARS-CoV-2 is repeatedly introduced, which is the case in most places and not all that surprising, but it would be great to convey just how important (or unimportant) these introductions were in driving the total case load.

In more details:

- There is a strong focus on variants of SARS-CoV-2, but, to me, it is unclear if the variants in anyway impacted the local transmission dynamics. I think if there is such a strong focus on variants, the connection to shaping the local dynamics should be made more explicit. Also, it is unclear from the manuscript if shifts in local frequencies of variants are due to changes in variant frequencies elsewhere or by chance, or whether these are due to differences in local transmission dynamics. Also (this is more minor), It would be great if the authors could elaborate at some place in the manuscript what they mean by variant (how are they defined, when do viruses that differ genetically belong to the same/different variant?).
- Many studies have shown that SARS-CoV-2 viruses are repeatedly introduced into different places. I think a more quantitative description of the importance of introductions would greatly benefit the manuscript. How much of the total cases in Rwanda are estimated to have been caused by introductions from elsewhere? What was the impact of those introductions, did they drive the different outbreaks? Also, in the analysis here, what is the benefit to including travel histories and genome sequences jointly? Do any of the results about from where SARS-CoV-2 was introduced change when including genome sequences vs. not including them?
- The authors note in the abstract that they discuss the potential impact of control measures on the spread of SARS-CoV-2. In my opinion though, there isn't really an analysis showing any impact of any measure on either introductions or local spread. It would be great to see an analysis that directly shows the impact of control measures, either on the local transmission rate or on the rate of importation.

Minor:

- Abstract: First sentence can be left out in my opinion. Starting with the second sentence is more motivating.
- Abstract: Test positivity is a measure that should not depend on variants, but merely the availability of test, local prevalence etc.
- Abstract: Considering that most vaccines, particularly the mRNA ones, show decent efficacy against the variants, I don't think that it has been shown that variants should affect vaccination efforts
- L109: There isn't really an analysis that infers/describes the transmission dynamics
- L112: Please elaborate on the benefit of using travel histories alongside genomes. What couldn't have been done just using both data sources independently.
- L124+: How does the biased sampling towards people sampled at the ports of entry affect the analyses/interpretation of what is going on
- L130+: As above, are these samples representative of what is going on?
- L146: Please elaborate on how this is its own variant.
- L150: Is this diversity representative of local spread or of what happened elsewhere during this time
- L152: Is this driven by a variant or simply by a growth of cases that were present at the time?
- L167: sentence unclear
- L181: introductions
- L193: Could this statement not also be made by just knowing that there were COVID cases that traveled from Burundi and South Sudan?
- L196+: I would stress that these are observed number of introductions and are just a lower bound

for the actual number of introductions

- L207: Does this require a phylogenetic analysis beyond just knowing which variant the case of a traveled person was infected with?
- L213+: Did this reveal routes of introductions beyond the ones identified from travel histories?
- L228: Please elaborate on how are you able to distinguish between local spread and separate introductions?
- L237: How are unsampled locations accounted for in the phylogeographic reconstruction?
- L243: Is this not mainly due to conditioning on that history?
- L264: How do you distinguish in this analysis between different introductions into Rwanda?
- L398: What is a GTR+F+R8 model?
- L424: and to efficiently

RESPONSES TO THE REVIEWERS' COMMENTS

Reviewer #1 (Remarks to the Authors)

> The authors look at the COVID-19 epidemic in Rwanda using one year of data. The study is certainly interesting and methods are cutting-edge. However, it presents a limitation: sampling bias. Looking at figure 1, the sampling bias appears quite striking: most of the sampling is coming from one region (name the region in the figure). I would like to see from the authors a supplementary figure that shows cases per region per month of sampling and corresponding breakdown of sampling per region (like panels a & b but for each month of sampling).

***Authors' response:** We thank the reviewer for the appreciation of our work and our use of recently developed methods to study the epidemic in Rwanda. We agree that the current genome surveillance efforts mostly focused on the capital region and now provide the requested Supplementary Figure S1 showing the cases per region and per month of sampling. We would also like to point out that this within-country sampling bias does **not** introduce sampling bias in our country-level phylogeographic analysis shown in Figures 5 and 6 in the main text. Further down this point-by-point response, where the Reviewer refers to the corresponding Figure S5 (now Supplementary Figure S9), we respond in detail regarding sampling bias and its potential impact on the continuous phylogeographic inference we presented in Supplementary Materials. As per the Reviewer's request, we have now added a new Supplementary Figure S1 that shows the cases per region and per month of sampling. Figure S1 shows that Kigali consistently reported the majority of cases, but also that from the end of 2020 onward cases started appearing throughout the entire country. Whereas the genomes in our study were mostly collected from Kigali, we did also make use of those sequences from the other provinces in our analyses.*

> Moreover, there are few points during which cases were decreasing or not surpassing 200 counts and for which the authors have many sequences, but when the epidemic started accumulating more cases (jan2021 and after), sampling was scarce (23 and 5 sequences, with 5 sequences coming from a time when daily new cases were almost 600). Please add: "daily cases" to y axis in fig 1c so that the reader can easily interpret the figure without reading the legend.

***Authors' response:** We have made the requested modification to Figure 1 (and changed the order of the subfigures) and now more clearly mention these differences between case counts and sampled genomes in the text.*

> Wouldn't the bias in sequences from Kigali bias the analysis reported in figure S5? What did the authors do to prevent such a potential issue?

***Authors' response:** The Reviewer raises an important issue regarding the within-country phylogeographic analysis. As already acknowledged in the text, the relatively higher sampling effort in Kigali indeed impacts the continuous phylogeographic reconstruction in the sense that this analysis likely fails to highlight local circulation of lineages outside the capital city as a*

consequence of under-sampling in those regions. Despite this limitation, which is inherent to this important sampling effort within Rwanda, we aimed to perform a continuous phylogeographic analysis to investigate the evolutionary relationships of our samples in a spatio-temporal context. While the heterogeneous sampling effort (or sampling bias) indeed prevents us from interpreting this reconstruction as a realistic overview of the overall dispersal history of SARS-CoV-2 lineages in Rwanda, it still aims to infer the dispersal history of those lineages that were sampled in our study. As previously discussed in Dellicour et al. (2019, doi: 10.1111/mec.15222): although a particular sampling effort will always affect the reconstructed dispersal history of viral lineages, continuous phylogeographic inference will still provide movement data that can inform on the dispersal dynamics of the virus. These aspects are now explicitly acknowledged in the Supplementary Text. Furthermore, we also explicitly mention as a further perspective that investigating the local dispersal dynamic of viral lineages outside Kigali would require a more intensive sampling effort outside the capital city, which is currently not available.

To address the concern raised by the Reviewer, we have now performed a sensitivity test by redoing our continuous phylogeographic analysis on ten subsampled data sets with reduced heterogeneous sampling by downsampling the more intensively sampled areas. To this end, we randomly selected a maximum of two sequences per administrative “sector” area in each of the ten replicate data sets. We have added Supplementary Text as well as a new Supplementary Figure S10 that shows the resulting phylogeographic reconstructions to be coherent with the phylogeographic pattern inferred from the full data set.

> Please make sure to describe in more detail what figures 2 and 3 indicate, otherwise move them to supplementary. Right now, it seems like the reader has to interpret them since they are only briefly mentioned in line 135.

Authors’ response: *We thank the Reviewer for this suggestion and now describe both Figures 2 and 3 in more detail in the main text.*

> Shouldn’t paragraph “Phylogenetic analysis” called instead “lineage characterization”?

Authors’ response: *We have renamed the paragraph as suggested by the Reviewer.*

> Lines 213-215 refer to bayes factors, not markov jumps, therefore the reference to figure 7 is not correct. Markov jumps are described afterwards and that’s where the ref to figure 7 is needed. I think figure S3 should not be supplementary.

Authors’ response: *As per the Reviewer’s request, we have now moved Supplementary Figure S3 into the main text (as Figure 8). We thank the Reviewer for pointing out the inconsistency in our writeup and have removed the reference to Figure 7 in that sentence.*

> There is no reference to Figure S1 in the text.

Authors' response: *We apologize for this oversight and now refer to Supplementary Figure S1 (now Supplementary Figure S3) in the main text.*

> Authors need to add an GISAID author acknowledgement table in supplementary with all the strains that were used from GISAID.

Authors' response: *We apologize for this oversight and have now added a GISAID author acknowledgement table as supplementary material.*

> Besides the above mentioned limitation, the manuscript is poorly formatted. Some figures are not discussed in the text or mentioned, and some others that might be more relevant are found in the supplementary materials rather than in the main text.

Authors' response: *We apologize for these issues and have modified various parts of the manuscript to ensure that all figures and tables are properly discussed and referred to in the main text. As per the Reviewer's request, we have moved Supplementary Figure S3 into the main text (now Figure 8), along with a description of these results.*

Reviewer #2 (Remarks to the Authors)

> In this manuscript, the authors describe a study, where they sequenced over 200 cases of SARS-CoV-2 in Rwanda. Using these sequences, they then describe the importance of introductions that have led to these cases. Using phylogenetic reconstruction, alongside travel information of sampled individuals, they reconstruct the phylogenetic tree of sequences from Rwanda alongside other sequences from the region and around the globe to determine the spatial spread of SARS-CoV-2 in Rwanda. Additionally, they find an abundance of several different Variants of SARS-CoV-2 as well as such that are designated variants of concern. The manuscript importantly fills a gap in the current literatures of SARS-CoV-2 by analyzing its spread in a region that has been under covered elsewhere and the manuscript is overall well written. I do though think though that the manuscript would greatly profit by going beyond mostly standard phylogenetic and ancestral reconstructions and largely descriptive analyses, to be a bit more quantitative. For example, there are different variants, but did they affect anything or does the shift in variants mainly reflect what is happening elsewhere and how are the shifts in variants not just by chance?

Authors' response: *We thank the Reviewer for this suggestion but want to first point out that our phylogeographic reconstructions were performed according to the latest developments in the field, as acknowledged by the first Reviewer. Accommodating individual travel histories in Bayesian phylogeographic inference is a very recent development, as it was published in October 2020 by Lemey et al., with several of the authors of this manuscript contributing to its*

development. As such, our current study on SARS-CoV-2 in Rwanda is among the first to make use of this novel phylogeographic inference approach.

Regarding the different lineages that are the focus of our study (A.23.1 and B.1.380), we now discuss the epidemic situation in Rwanda, Uganda and Kenya in more detail in Supplementary Materials. To this end, we provide new visualizations (Supplementary Figures S6 - S8) on top of an additional test in the main manuscript (Bayesian Tip-association Significance testing; BaTS) which enabled us to formally determine that the A.23.1 and B.1.380 epidemics in Rwanda were driven by local transmission, more so than by introductions. We have refrained from delving into follow-up quantitative analyses (or formal testing) in terms of comparing the number of introductions - as estimated through the reported Markov jumps - with the size of local transmission clusters, as the limited number of genomes from Rwanda and - perhaps more importantly - its neighboring / surrounding countries will not permit a fair assessment. We also list similar studies to our own in which such quantitative analyses were not performed but instead discussed in a more descriptive manner as well, arguably for the same reasons as those listed here.

In short, we now discuss that the B.1.380 epidemic was not seen in Rwanda's neighboring / surrounding countries and that within Rwanda this epidemic was responsible for the majority of cases reported throughout the first lockdown (March - July 2020; borders were closed) and until the emergence of A.23.1. When this latter lineage emerged in Rwanda and Uganda, it became responsible for the increase in cases in both countries (Bugembe et al., 2021). This was not the case for Kenya however, where A.23.1 has only played a minor role so far, while appearing to increase in frequency towards the end of our study period.

> Also, the authors show that SARS-CoV-2 is repeatedly introduced, which is the case in most places and not all that surprising, but it would be great to convey just how important (or unimportant) these introductions were in driving the total case load. In more details: There is a strong focus on variants of SARS-CoV-2, but, to me, it is unclear if the variants in any way impacted the local transmission dynamics. I think if there is such a strong focus on variants, the connection to shaping the local dynamics should be made more explicit. Also, it is unclear from the manuscript if shifts in local frequencies of variants are due to changes in variant frequencies elsewhere or by chance, or whether these are due to differences in local transmission dynamics.

Authors' response: *The Reviewer is correct in that the A.23.1 and B.1.380 lineages were introduced into Rwanda from abroad as we have indeed shown in our work, and that this pattern can be seen in many countries. However, our phylogeographic analyses as well as the BaTS analyses, we have now performed point to mostly local transmission, after a relatively limited number of introductions, driving the epidemic in Rwanda. This is in contrast with how the epidemic saw increases in case counts in other countries, such as Belgium for example (Dellicour et al., 2021), which we now mention in Supplementary Materials, along with other published studies.*

To obtain better insights into possible shifts in local frequencies of the different variants circulating, we have created custom lineage frequency plots which enabled comparing the epidemic situation in Rwanda to those of two neighboring countries for which a rather similar number of genomes were available. As we now discuss in detail, the B.1.380 epidemic was

specific to Rwanda and did not occur in neither Uganda nor Kenya (other surrounding countries did not have a reasonable number of available genomes to be added to the comparison). In terms of the A.23.1 epidemic, similarities - albeit with a clear shift in time - can be seen between Rwanda and Uganda, with this lineage however not being of much importance in Kenya during our study period. Based on our observations that these frequency plots can be sensitive to low numbers of cases attributed to certain lineages, we have opted to put this information in Supplementary Materials and point out this issue.

> Also (this is more minor), It would be great if the authors could elaborate at some place in the manuscript what they mean by variant (how are they defined, when do viruses that differ genetically belong to the same/different variant?).

Authors' response: *The Reviewer raises a valid point in that we did not explain this difference in definition. We have now added this to the introduction.*

> Many studies have shown that SARS-CoV-2 viruses are repeatedly introduced into different places. I think a more quantitative description of the importance of introductions would greatly benefit the manuscript. How much of the total cases in Rwanda are estimated to have been caused by introductions from elsewhere? What was the impact of those introductions, did they drive the different outbreaks?

Authors' response: *As mentioned above, we have performed an additional BaTS analysis to determine whether introductions or local transmission are key in shaping the epidemic in Rwanda. BaTS yielded significant p-values for both A.23.1 and B.1.380 analyses, which indicate that local transmission chains have played an important role in driving the Rwandan epidemic, more so than introduction events into the country. Additionally, the introductions of A.23.1 and B.1.380 were indeed important as they rose to being dominant lineages (both in Rwanda, the former also in Uganda) that drove the increase in case counts. However, quantifying in more detail, i.e. beyond the result of the BaTS analyses we performed and in terms of coming up with actual numbers, would be too far-fetched in our opinion given the rather limited genome sequencing efforts of the countries involved and the complete lack of genome sequences from some of Rwanda's neighboring countries. We hence prefer to be cautious and not make such claims, which were also not made in the publications on SARS-CoV-2 in Uganda and Kenya for example. We discuss this in more detail in Supplementary Materials.*

>Also, in the analysis here, what is the benefit to including travel histories and genome sequences jointly? Do any of the results about from where SARS-CoV-2 was introduced change when including genome sequences vs. not including them?

Authors' response: *Including travel histories in phylogeographic inference enables more realistic phylogeographic inference to be performed, as shown in the original travel history publication (Lemey et al., 2020). There are several reasons for this, such as the possibility of including unsampled locations (countries in our analyses) so that these locations can be*

considered for the estimation of the ancestral locations, which would otherwise not be possible. Additionally, rather than estimating certain introduction events, we are able to treat documented travel histories as observed data so that these introduction events will with certainty be found in the resulting phylogeographic analysis. The Reviewer is correct in that we did not perform an additional (standard) phylogeographic analysis to compare the results in our manuscript with. However, given that the presence of lineages in unsampled locations is particularly problematic for the inferences in our manuscript, the use of the travel history-aware methodology is well justified over using standard discrete phylogeographic inference, which would be more prone to sampling bias than the approach we used.

> The authors note in the abstract that they discuss the potential impact of control measures on the spread of SARS-CoV-2. In my opinion though, there isn't really an analysis showing any impact of any measure on either introductions or local spread. It would be great to see an analysis that directly shows the impact of control measures, either on the local transmission rate or on the rate of importation.

***Authors' response:** We agree with the Reviewer that we did not establish a direct correlation of the spread of SARS-CoV-2 and the control measures, and we have hence modified this statement in the abstract. However, as demonstrated by our updated timeline Figure 1A of control measures, spikes in cases were associated with more lax control measures, whereas instances with stricter lockdown were associated with lower positive cases.*

> **Minor comments**

- Abstract: First sentence can be left out in my opinion. Starting with the second sentence is more motivating.

***Authors' response:** Done.*

- Abstract: Test positivity is a measure that should not depend on variants, but merely the availability of test, local prevalence etc.

***Authors' response:** The 'test positivity' was removed from the second sentence of the abstract.*

- Abstract: Considering that most vaccines, particularly the mRNA ones, show decent efficacy against the variants, I don't think that it has been shown that variants should affect vaccination efforts.

***Authors' response:** We agree with the reviewer that mRNA vaccines have demonstrated a decent efficacy against certain variants. However, a number of published studies have demonstrated an impaired and diminished immune response induced by vaccines for some variants of concern. For example, Becker et al. (Nature Communications 12:3109, 2021) reported a 'substantially*

reduced Ab neutralization for the B.1.351 variant' on sera obtained from vaccinated people, highlighting the importance of genomic surveillance and further vaccine efficacy studies in different settings. We have incorporated this information into the Discussion section of our manuscript.

- L109: There isn't really an analysis that infers/describes the transmission dynamics

Authors' response: *We have now rewritten this sentence.*

- L112: Please elaborate on the benefit of using travel histories alongside genomes. What couldn't have been done just using both data sources independently.

Authors' response: *We have extended the description of this approach in the 'Phylogeographic reconstruction accommodating individual travel histories' section. In summary, these travel histories constitute additional observed data that we use to inform the discrete phylogeographic analyses in our manuscript, for both lineages A.23.1 and B.1.380, using a recently developed methodology (Lemey et al., 2020). Performing joint inference on the genome and travel data enables considering countries from which no genomes are available as ancestral locations of the spread of these lineages, as well as informing the spread between locations by using known travel events (instead of otherwise estimating such transitions between locations without knowing for sure if there had been any such events).*

- L124+: How does the biased sampling towards people sampled at the ports of entry affect the analyses/interpretation of what is going on?

Authors' response: *We agree with the Reviewer on the presence of sampling bias in our data set, an issue also raised by the first Reviewer. As discussed in reply to the comment of the first Reviewer and to assess the relevance of this bias with respect to our reported results, we have performed additional subsampling analyses in the Supplementary Materials and show that these subsampled phylogeographic reconstructions are coherent with the phylogeographic pattern inferred from the original / full data set in our manuscript.*

- L130+: As above, are these samples representative of what is going on?

Authors' response: *The information listed in this section constitutes the observed travel history data which we will use to perform our phylogeographic analyses for lineage A.23.1 and B.1.380. While these are not all travel cases into Rwanda, they are the ones for which we have genomic data available. Unfortunately, we do not currently have sufficiently detailed information for the incoming travelers via land but now provide this information for the incoming travelers via air in the caption of Supplementary Figure S2. While not attaining perfect proportionality, our included travel cases are representative of the incoming travelers via air. Finally, for the discrete phylogeographic reconstructions, it is advised to make use of as much travel history data as possible, to inform these analyses as best as possible, which is the approach we have*

taken.

- L146: Please elaborate on how this is its own variant.

Authors' response: *We consider B.1.380 to be its own lineage, as determined through the Pangolin module, but not a variant. This is discussed in the methods section, in the subsection 'Phylogenetic and phylogeographic analysis'.*

- L150: Is this diversity representative of local spread or of what happened elsewhere during this time

Authors' response: *We now compare in Supplementary Materials the diversity of lineages in Rwanda to that of two of its surrounding countries that have released a similar number of SARS-CoV-2 genomes. We show that, while each country has its own dynamics, Rwanda and Uganda have seen a similar rise in number of infections with lineage A.23.1, whereas the surge in infections with B.1.380 was specific to Rwanda.*

- L152: Is this driven by a variant or simply by a growth of cases that were present at the time?

Authors' response: *As we now discuss in Supplementary Materials, the increase in case counts coincides with the frequency shift caused by lineage A.23.1, which rose to dominance starting at the end of 2020.*

- L167: sentence unclear

Authors' response: *The sentence was rephrased for more clarity to: "Importantly, a recent study suggests that B.1.1.7 is not only more transmissible than preexisting SARS-CoV-2 variants, but that it may also cause more severe illness and is associated with increased mortality."*

- L181: introductions

Authors' response: *We have modified 'introduction' to "introductions".*

- L193: Could this statement not also be made by just knowing that there were COVID cases that traveled from Burundi and South Sudan?

Authors' response: *We prefer to perform, where possible and appropriate, state-of-the-art phylogenetic and phylogeographic approaches to estimate the origin and ensuing viral dispersal of SARS-CoV-2 lineages. Such approaches enable us to look into the evolutionary and geographical history of these lineages, which descriptive summaries of travel cases would not enable us to do.*

• L196+: I would stress that these are observed number of introductions and are just a lower bound for the actual number of introductions

Authors' response: We had already mentioned that these estimates represent minimum numbers (i.e. lower bounds) but have added this again in the next part of the sentence.

• L207: Does this require a phylogenetic analysis beyond just knowing which variant the case of a traveled person was infected with?

Authors' response: The Reviewer is correct and we have now rewritten this sentence to acknowledge this.

• L213+: Did this reveal routes of introductions beyond the ones identified from travel histories?

Authors' response: The introductions identified from individual travel histories can be seen at the tips of the phylogenies shown in Figures 5 and 6. However, all other transitions in those figures, i.e. between ancestral nodes in those phylogenies, are inferred through phylogeographic inference and could not be obtained by inspecting the documented travel cases alone.

• L228: Please elaborate on how you are able to distinguish between local spread and separate introductions?

Authors' response: Both can be observed in the MCC trees shown in Figures 5 and 6, where a transition between internal nodes estimated to lie in different countries are considered to be introductions whereas clades of nodes estimated to lie in the same country are considered to represent local spread. When support for the inferred ancestral nodes may at times be low, the distinction may be less clear however, which is why the use of estimates such as Markov jumps and analyses such as BaTS offer a way to maintain a proper overall view on the epidemic in Rwanda.

• L237: How are unsampled locations accounted for in the phylogeographic reconstruction?

Authors' response: As we have indicated in the manuscript, we have made use of a recent development in discrete Bayesian phylogeographic inference (see Lemey et al., 2020). The possibility to include unsampled locations is hence not a novel part of our manuscript, and we refer to Lemey et al. (2020) for more information.

• L243: Is this not mainly due to conditioning on that history?

Authors' response: No, we only treat the introductions from Tanzania into Rwanda for this travel case as observed data. Any events prior to that are inferred through phylogeographic inference and summarized from the posterior tree distribution, thereby incorporating phylogenetic uncertainty.

• L264: How do you distinguish in this analysis between different introductions into Rwanda?

Authors' response: The continuous phylogeographic reconstruction discussed here focuses on reconstructing the viral spread within Rwanda. To this end, we extracted from the overall MCC

tree the largest possible clades that only had Rwandan genomes as their tips. In other words, the data selection for this analysis is based on a simple tip-coloring approach of the ML tree shown in (now) Supplementary Figure S3.

- L398: What is a GTR+F+R8 model?

Authors' response: *We now explain this abbreviation in more detail in the text: 'A general time-reversible model with empirical base frequencies and an auto-discrete-gamma model for rates over sites with 8 rate categories (GTR+F+R8).'*

- L424: and to efficiently

Authors' response: *We have made the requested modification as suggested by the Reviewer.*

REVIEWERS' COMMENTS

Reviewer #1 (Remarks to the Author):

The authors addressed all my comments. I have no further comments.

Reviewer #2 (Remarks to the Author):

Most of my comments have been addressed, though I would add one supplementary figure to show if doing a phylogeographic reconstruction with travel histories adds any more information about from where introductions came from than just using the travel histories. Other than that, I only have some minor comments left.

Medium:

- If the authors don't want to compare if the addition of travel histories changes the introduction patterns compared not in including them in the phylogeographic reconstruction, that is reasonable. It would be great thought to replicate the markov jump figure in the supplement by counting the number of introductions for the two variants by only counting travel histories alone, without doing any phylogenetic reconstruction. I.e just count the number of genomes from subtree A and subtree B1 with travel history to Uganda, Kenya, Italy etc. to see if the phylogenetic reconstruction adds anything more than what is captured by the travel histories.

Minor:

- The analyses about how important local spread is with BATS only allows to quantify whether there is some significant local structuring, but not really how important the structure is, so I would rephrase the sentence "local transmission chains have played an important role in driving the Rwandan epidemic, more so than introduction events into the country." to reflect this. Also, I don't think this analysis really answers how important introductions are (only that there is geographical clustering) and can also be left out or moved to the supplement.
- For the continuous phylogeographic analysis, it seems like the result will always be (as for any diffusion model) that there is radiation out from some point. Is there any other potential result from the analysis with that model than spread from Kigali to the rest of the country? If that is essentially the prior assumption of the continuous diffusion model, then this should be stated.
- However, because of the differences in sequencing..... sentence needs some rephrasing
- For this site model, I would use the designation GTR+Gamma_4 with empirical frequencies instead, as this is more common.

RESPONSE TO REVIEWERS' COMMENTS

Reviewer #1 (Remarks to the Author):

The authors addressed all my comments. I have no further comments.

Response: We thank the Reviewer for this positive assessment.

Reviewer #2 (Remarks to the Author):

Most of my comments have been addressed, though I would add one supplementary figure to show if doing a phylogeographic reconstruction with travel histories adds any more information about from where introductions came from than just using the travel histories. Other than that, I only have some minor comments left.

Medium:

- If the authors don't want to compare if the addition of travel histories changes the introduction patterns compared not in including them in the phylogeographic reconstruction, that is reasonable. It would be great thought to replicate the markov jump figure in the supplement by counting the number of introductions for the two variants by only counting travel histories alone, without doing any phylogenetic reconstruction. I.e just count the number of genomes from subtree A and subtree B1 with travel history to Uganda, Kenya, Italy etc. to see if the phylogenetic reconstruction adds anything more than what is captured by the travel histories.

Response: We thank the Reviewer for this suggestion, as performing an explicit comparison is very time-consuming and the publication that details the travel

history-aware phylogeographic reconstruction approach has already shown the benefits of including individual travel histories on two different data sets by performing such time-consuming explicit comparisons (Lemey et al., 2020).

We have now added in Supplementary Materials the suggested comparison by the Reviewer on both subtrees, showing that the phylogeographic reconstruction does not merely report the individual travel histories but shows clear differences - in both directions - between the estimated number of Markov jumps and the number of recorded travel histories, in part because of the conceptual difference between these two ways of recording transitions between locations (e.g. an infected family of four crossing a country border would result in four individual travel histories but only in a single Markov jump, on the assumption that they have a single source of infection). The figures that hold these comparisons are shown below and are part of a newly added section in Supplementary: “A note on exploiting individual travel histories through phylogeographic reconstruction”.

Supplementary Figure S9. Subtree A introductions into Rwanda by country of origin and travel cases. The posterior mean number of Markov jumps obtained from the phylogeographic analysis for lineage A.23.1 is plotted against the number of travel-associated cases from each location. Only countries of origin

with supported rates (Bayes factor >3) are shown. The phylogeographic analysis is able to identify multiple introductions from Uganda and Kenya not accounted for in the available travel records. However, in the case of Tanzania, the phylogeographic reconstruction shows a lower inferred number of introduction events (2.6) than the number of recorded travel cases (4). This can be explained by the fact that the Tanzanian sequences with associated travel dates are clustered in two pairs with identical sequences and sampling dates.

Supplementary Figure S10. Subtree B.1 introductions into Rwanda by country of origin and travel cases. The posterior mean number of Markov jumps obtained from the phylogeographic analysis for lineage B.1.380 is plotted against the number of travel associated cases from each location. Only countries of origin with supported rates (Bayes factor >3) are shown. The phylogeographic analysis is able to identify multiple introductions from Kenya not accounted for in the travel cases available.

Minor:

- The analyses about how important local spread is with BATS only allows to quantify whether there is some significant local structuring, but not really how important the structure is, so I would rephrase the sentence “local transmission chains have played an important role in driving the Rwandan epidemic, more so

than introduction events into the country.” to reflect this. Also, I don’t think this analysis really answers how important introductions are (only that there is geographical clustering) and can also be left out or moved to the supplement.

Response: We have omitted the final part of this sentence, which now reads: “The significant degree of clustering suggests that for both of these lineages, local transmission chains have played an important role in driving the Rwandan epidemic.”

- For the continuous phylogeographic analysis, it seems like the result will always be (as for any diffusion model) that there is radiation out from some point. Is there any other potential result from the analysis with that model than spread from Kigali to the rest of the country? If that is essentially the prior assumption of the continuous diffusion model, then this should be stated.

Response: This is not a prior assumption of the continuous diffusion model implemented in BEAST 1.10, which we used to perform the continuous phylogeographic reconstruction. This model employs an anisotropic random walk diffusion process (Pybus et al. 2012; <https://doi.org/10.1073/pnas.1206598109>), which has demonstrated the ability to infer non-central locations of internal nodes. Additional examples can be found in Kalkauskas et al. (2021; e.g. Figure 6; <https://doi.org/10.1371/journal.pcbi.1008561>).

- However, because of the differences in sequencing..... sentence needs some rephrasing

Response: We have rewritten this sentence to “However, because of the differences in sequencing efforts across the globe, we cannot dismiss the possibility of intermediary locations in these cases”.

- For this site model, I would use the designation GTR+Gamma_4 with empirical frequencies instead, as this is more common.

Response: While we agree that the GTR+G4 model is more widely used, the model we ended up using (i.e., GTR+F+R8) was determined through the automated model selection procedure in IQ-TREE and as such yielded a better fit to the data than GTR+G4. We hence cannot merely alter the name of the model as this would not correspond to the analysis that was performed (and that was objectively determined through a model selection procedure).